# Stochastic Modelling of ^13^C NMR Spin Relaxation Experiments in Oligosaccharides

**DOI:** 10.3390/molecules26092418

**Published:** 2021-04-21

**Authors:** Sergio Rampino, Mirco Zerbetto, Antonino Polimeno

**Affiliations:** 1Scuola Normale Superiore, 56126 Pisa, Italy; sergio.rampino@sns.it; 2Dipartimento di Scienze Chimiche, Università degli Studi di Padova, 35131 Padova, Italy; mirco.zerbetto@unipd.it

**Keywords:** NMR spin relaxation, stochastic modelling, oligosaccharides

## Abstract

A framework for the stochastic description of relaxation processes in flexible macromolecules including dissipative effects has been recently introduced, starting from an atomistic view, describing the joint relaxation of internal coordinates and global degrees of freedom, and depending on parameters recoverable from classic force fields (energetics) and medium modelling at the continuum level (friction tensors). The new approach provides a rational context for the interpretation of magnetic resonance relaxation experiments. In its simplest formulation, the semi-flexible Brownian (SFB) model has been until now shown to reproduce correctly correlation functions and spectral densities related to orientational properties obtained by direct molecular dynamics simulations of peptides. Here, for the first time, we applied directly the SFB approach to the practical evaluation of high-quality ^13^C nuclear magnetic resonance relaxation parameters, $T_{1}$ and $T_{2}$, and the heteronuclear NOE of several oligosaccharides, which were previously interpreted on the basis of refined ad hoc modelling. The calculated NMR relaxation parameters were in agreement with the experimental data, showing that this general approach can be applied to diverse classes of molecular systems, with the minimal usage of adjustable parameters.

## 1. Introduction

Monitoring and describing molecular dynamics are an important area of investigation in modern physical chemistry. Internal and global motions in solution affect directly or indirectly most spectroscopic methods aimed at the characterization of non-rigid molecules such as Nuclear Magnetic Resonance (NMR) relaxation [1,2,3], fluorescence anisotropy decay [4], time-resolved X-ray [5] and in single-molecule experiments such as site-directed spin-labelled electron spin resonance [6,7], Förster fluorescence resonance energy transfer [8] and atomic force microscopy [9].

In particular, Nuclear Magnetic Resonance (NMR) spectroscopy is known to be an important and powerful experimental technique for the observation of the dynamic properties of macromolecules. Some of the macroscopic physical observables are the relaxation times $T_{1}$, $T_{2}$ and the heteronuclear Overhauser Effect (NOE) of ^15^N, ^2^H and ^13^C nuclei, which are extremely sensitive to molecular motions, leading to the possibility to understand localized dynamics (e.g., studying conformational motions specifically in the active site of a protein) and to build a spatially distributed map of the macromolecule flexibility. However, interpretative tools can be complex due to several factors such as (i) the necessity to take into account diverse kinds of interactions, e.g., dipolar ^15^N and ^13^C and quadrupolar ^2^H interactions, (ii) the coupling between global reorientation and large amplitude motions of entire domains, as well as limited local readjustments and restricted single-residue motions. In general, different spectroscopic techniques probe different physical observables, which, in addition, provide information on motions taking place at different time-scales. It seems therefore particularly important to introduce relevant sets of coordinates that are adapted to the observable involved in a particular experimental approach. This consideration is especially relevant in the case of NMR relaxation [1,10], for which interpretative methods for internal relaxation processes were introduced early on in the form of adaptable simple spectral densities, as in the Lipari–Szabo (LS) approach [11,12], or later, in the form of explicit dynamic models, as for instance in the Slowly Relaxing Local Structure (SRLS) model [13,14].

A rational approach to the in silico interpretation of the relaxation data of flexible molecules needs two distinct elements. First, a precise geometrical analysis is needed in order to relate properly the dynamic model to a set of experimentally observable quantities. In particular, care should be taken to account for the tensorial nature of the spectroscopic interactions, by defining proper local frames of reference. Next, the relaxation times or other observables are linked to time correlation functions/spectral densities of a specific nature that can be evaluated on the basis of the dynamic model itself. The latter can range from a full atomistic molecular dynamics (MD) simulation-based approach to simplified semi-analytical expressions for the correlation functions. At an intermediate level of complexity, several approaches have been devised based on various approximations. For instance, one can make the simplifying assumption that local motions are due, at least for semi-rigid systems, to a network of dynamically coupled neighbours (network model) [15,16] or caused by partial diffusive reorientation within a local potential (SRLS) [14]. One can also assume specific statistical characteristics (diffusive or Brownian dynamics, fractional Brownian dynamics [17], etc.).

Recently, a systematic approach [18,19] has been proposed that tries to combine a detailed definition of the molecular geometry and a correct description of the associated dynamical features. The method attempts to include the information on the molecule geometry, topology and interactions into a general stochastic model, which can be tailored at different levels of accuracy introducing specific approximations based on time-scale separation arguments. A master equation can be obtained and, with suitable approximations, numerically solved. In particular, a basic implementation, named the Semi-Flexible Brownian (SFB) model, has been developed for the description of partially flexible macromolecules in solution. Until now, the SFB model has been applied only to model cases, and no examples have been shown of calculations of directly measurable observables. In this paper, we present a full investigation of the SFB performance for the evaluation of ^13^C nuclear magnetic resonance relaxation parameters, $T_{1}$ and $T_{2}$, and the heteronuclear NOE of several oligosaccharides, which were previously interpreted on the basis of ad hoc stochastic modelling. In particular, we discuss the computational strategy and implementation of the method and detailed results, which confirm how the calculated NMR relaxation parameters are in satisfactory agreement with the experimental data, and we suggest that this general approach can be safely applied to diverse classes of molecular systems, with a minimal usage of adjustable parameters.

The paper is organized as follows. Section 2 summarizes the basic features of the SFB model and its implementation. The main results are shown in Section 3. A discussion is provided in Section 4.

## 2. Methods

### 2.1. Observable and Geometric Setup

A spectroscopic observable is written usually in terms of suitable time correlation functions or spectral densities, and their fast and accurate evaluation are the main objective of a dynamic modelling approach. The distinction between the description of the dynamics of the molecular system and the definition of the physico-chemical observable is often skipped over. We review here some salient points useful for the general comprehension of the topic.

Magnetic relaxation times $T_{1}$, $T_{2}$ and the NOE of ^15^N, ^13^C and ^2^H nuclei depend on dipolar (^15^N and ^13^C) and quadrupolar (^2^H) interactions, on chemical shift anisotropy and cross-correlation effects. In general, we define the following set of reference frames: (i) a Laboratory Frame (LF), i.e., a fixed external frame; (ii) an “Attached” Frame (AF), i.e., a frame attached to the molecule, where the exact way of defining the AF is actually model-dependent and is left temporarily undefined, noting that the choice of the AF while straightforward for a rigid molecule is not trivial for a flexible system; (iii) an interaction frame (μF), i.e., a local frame linked to the AF where some specific second-rank tensor spectroscopic property μ is well represented. Depending on the problem at hand, this could be for instance the frame where the ^13^C-^1^H dipolar or ^13^C chemical shift (CSA) tensors are diagonal [20,21]. In Figure 1, an example is shown of the frame choice to compute the NMR relaxation data of a ^13^C-^1^H probe. The set of Euler angles Ω or other orientational coordinates transforming from the Laboratory Frame (LF) to the AF is time dependent and linked to the local restricted motions, large amplitude conformational motion and global orientation of the molecule. The dipolar and CSA frames are usually supposed to be rigidly attached to the AF. Here, only the Dipolar Frame (DF) is shown as an example.

Let us briefly describe the evaluation of the dipolar contribution, as an example, to clarify the relation between the dynamical model and the physical observable. As is shown in Appendix A, the NMR relaxation observables are functions of the spectral densities of the correlation functions: (1)$$G_{k,k^{\prime }}\left(t\right)=C\bar{D_{k,0}^{2\;\;*}\left(\Omega_{D}\left(0\right)\right)D_{k^{\prime },0}^{2}\left(\Omega_{D}\left(t\right)\right)}$$

Here, *C* is a constant depending on the interaction type and geometry (cf. Appendix A). Using the properties of Wigner rotation matrices [22], the correlation functions can be rewritten as: (2)$$G_{k,k^{\prime }}\left(t\right)=C\sum_{m,m^{\prime }=-2}^{2}D_{m,0}^{2\;\;*}\left(\Omega_{\mathrm{AD}}\right)D_{m^{\prime },0}^{2}\left(\Omega_{\mathrm{AD}}\right)\bar{D_{k,m}^{2\;\;*}\left(\Omega (0)\right)D_{k^{\prime },m^{\prime }}^{2}\left(\Omega (t)\right)}$$

Here, $\Omega_{\mathrm{AD}}=\left(0,\pi /2,0\right)$. Thus, the expression for the correlation functions simplifies to: (3)$$G_{k,k^{\prime }}\left(t\right)=C\sum_{m,m^{\prime }=-2}^{2}d_{m,0}^{2}\left(\pi /2\right)d_{m^{\prime },0}^{2}\left(\pi /2\right)\bar{D_{k,m}^{2\;\;*}\left(\Omega (0)\right)D_{k^{\prime },m^{\prime }}^{2}\left(\Omega (t)\right)}$$
with $d_{0,0}^{2}\left(\pi /2\right)=-1/2$, $d_{\pm 1,0}^{2}\left(\pi /2\right)=0$ and $d_{\pm 2,0}^{2}\left(\pi /2\right)=\sqrt{3/8}$.

The spectral densities for the dipolar interaction are thus calculated as:(4)$$J_{k,k^{\prime }}\left(\omega \right)=\int_{0}^{\infty }dte^{i\omega t}G_{k,k^{\prime }}\left(t\right)=C\sum_{m,m^{\prime }=-2}^{2}d_{m,0}^{2}\left(\pi /2\right)d_{m^{\prime },0}^{2}\left(\pi /2\right)j_{m,m^{\prime }}^{\left(k,k^{\prime }\right)}\left(\omega \right)$$

The spectral densities $j_{m,m^{\prime }}^{\left(k,k^{\prime }\right)}\left(\omega \right)$ are directly recoverable from the description of the molecular motion, which can be now discussed independently of the specific interaction, described at the desired level of accuracy (from an all-atom molecular dynamics treatment at one extreme to a description based on a rigid diffusive model at the opposite extreme).

### 2.2. Dynamic Model

In this work, we used the semi-flexible Brownian (SFB) model that was developed for the description of partially flexible macromolecules in solution, introduced in [18,19]. The SFB approach describes, in essence, the case of a flexible rotating molecule, assuming a generic energy function defined by harmonic coordinates and their conjugate momenta. The model neglects large-amplitude activated torsional kinetics and/or crankshaft motions [23,24], as well as second-order precession effects. Internal motions are described as a harmonic or boson bath, which retains full coupling with external motion and includes dissipative/stochastic effects. The SFB has been derived as a simple tool to describe molecular relaxation processes based on the Fokker–Planck (FP) [25] equations, which is suitable and computationally efficient even in problems of large dimensions and is solidly founded on structural information comparable to a standard molecular dynamics simulation, but amenable to a semi-analytical solution. Details of the model and the proposed methods for the numerical solution were presented in [19], and we limited ourselves here to a brief summary. The SFB model is based on the Fokker–Planck equation for the probability density ρ(Q,t): (5)$$\begin{array}{ccc} \frac{\partial \rho (Q,t)}{\partial t} & = & -\hat{\Gamma }\rho \left(Q,t\right)\\ \hat{\Gamma } & = & {\hat{\Gamma }}_{0}+{\hat{\Gamma }}_{\mathrm{int}}=-\sum_{i,j=1}^{N}\omega_{ij}^{\mathrm{io}}\frac{\partial }{\partial x_{i}}p\left(x\right)\frac{\partial }{\partial x_{j}}p{\left(x\right)}^{-1}+\sum_{i=1}^{N}\sum_{p=1}^{3}\omega {}_{ip}^{\mathrm{int}}x_{i}{\hat{M}}_{p} \end{array}$$

Here, Q=(Ω,x), where Ω denotes the set of Euler angles describing the orientation of the molecular frame AF (vide supra) with respect to the LF and $x=(x_{1},\ldots ,x_{i},\ldots ,x_{N})$ is a set of dimensionless harmonic degrees of freedom, obtained as linear combinations of internal coordinates, their conjugate momenta and external (angular) momentum components. ${\hat{M}}_{p}$, p=1,2,3, are the components of the infinitesimal rotation operator in the AF. As discussed in [18], this is the simplest description, recoverable from an initially atomistic model, of the Brownian dynamics of a non-rigid body, accounting for inertial effects and coupling between rotation and change of shape. Indeed, it describes the semi-rigid macromolecule of *n* atoms (or extended atoms, when a coarse-grained representation of the molecule is used), as a rotator coupled to N=6n−9 (i.e., 3n−6 internal coordinates, 3n−6 internal momenta and 3 components of the angular momentum L vector) harmonic degrees of freedom, in a fashion quite similar to standard spin-boson quantum mechanical approaches. Here, $p\left(x\right)=\exp\left(-x^{2}/2\right)/{\left(2\pi \right)}^{N/2}$ is the Gaussian distribution of the *N* modes, and the equilibrium distribution is $\rho \left(Q\right)=p\left(x\right)/8\pi^{2}$; $\omega_{ij}^{\mathrm{io}}$ is a (non-symmetric) matrix describing the relaxation of the internal coordinates, while $\omega {}_{ip}^{\mathrm{int}}$ dictates the coupling between rotation and internal coordinates. Both $\omega_{ij}^{\mathrm{io}}$ and $\omega {}_{ip}^{\mathrm{int}}$ can be derived from the molecular geometry and the generalized friction tensor obtained from a hydrodynamic description, which defines the dissipative forces acting on the molecule [18]. Approximate, but accurate semi-analytical solutions for correlation functions/spectral densities of interest, e.g., as given by Equation (Equation 4), have been presented based on extended perturbation treatments, which take advantage of the spin-boson structure of the time evolution operator to tackle the system dimensionality and devise a fast way of evaluating spectral densities of interest for the interpretation of nuclear magnetic resonance relaxation experiments [19].

### 2.3. Parameterization and Implementation

Matrix elements $\omega_{ij}^{\mathrm{io}}$ and $\omega {}_{ip}^{\mathrm{int}}$ are directly recoverable from an internal generic potential function defined with respect to natural coordinates, referred to as a local-minimum (reference) structure, and the related Hessian matrix, i.e., the Force Field (FF), as well as the friction tensor obtained for the local-minimum structure using a generalized hydrodynamic model [18,26]. Notice that more sophisticate choices are possible: a collection of reference molecular structures can be used, with or without the possibility of dynamical interconversion, the evaluation of internal energy directly from a short molecular dynamics simulation via a variance-covariance matrix evaluation for all the system internal coordinates and more refined approaches to evaluate dissipative properties, beyond the hydrodynamic limit. We focused on the most convenient choices from the computational point of view; the method was implemented in the form of an integrated package, the Stochastic Augmented Liouville Equation Method (SALEM), which operates from scratch reading the single reference structure, evaluating all the internal parameters based on some assumed FF and the basic macroscopic properties of the medium (e.g., viscosity) and estimating the relaxation parameters, based on the procedure summarized in Appendix A, through the calculation of the spectral densities defined in Equation (Equation 4). Following previous work [26], we allow as the only free parameter of the model the hydrodynamic average radius of atoms $R_{\mathrm{eff}}$, which is necessary for the calculation of the friction tensor.

Despite the high complexity of the basic theory, its application is relatively straightforward. As mentioned above, we planed to make SALEM made available soon as an open-access tool to the scientific community. Presently, the code is in its beta test phase of development, but it is already capable of (i) reading a PDB file together with very few other data, part of which comes from the experimental setup (temperature, viscosity, type of probe and its geometry, spectrometer frequency) and part from the physico-chemical properties (FF, hydrodynamic boundary conditions and effective radius), (ii) performing, currently invoking Tinker [27], a simple energy minimization, (iii) evaluating the friction tensor, (iv) defining the parameters contained in Equation (5), (v) evaluating the spectral densities defined in Equation (Equation 4), and finally, (vi) evaluating the relaxation times and NOEs. The computational times for the systems considered here went from a fraction of a second to a few minutes on a standard Nvidia GPU for gaming (larger molecules would require GPUs equipped with additional RAM memory and CUDA processes). Before fully publishing the code, at least in a pre-release phase useful for interested researchers, we intend to streamline the various operations under a general intuitive GUI and simplify the compilation procedures under common operating systems.

## 3. Results

We tested our method on a collection of oligosaccharides, for which previous analyses based on specific stochastic models were presented. The following systems are considered:α-L-Rha*p*-α-(1→2)-α-L-Rha*p*-OMe (two residues, **R2R**); experimental data: Reference [28]β-D-Glc*p*-(1→6)-α-D-[6-${}^{13}$C]-Man*p*-OMe (two residues, **BGL**); experimental data: Reference [29]β-D-Glc*p*-(1→3)[β-D-Glc*p*-(1→2)]-α-D-Man*p*-OMe (three residues, **GGM**); experimental data: Reference [30]α-D-Man*p*-(1→2)-α-D-Man*p*-(1→6)-α-D-[6-${}^{13}$C]-Man*p*-OMe (three residues, **TRI**); experimental data: Reference [31]α-L-Fuc*p*-(1→2)-β-D-Gal*p*-(1→3)-β-D-Glc*p*NAc-(1→3)-β-D-Gal*p*-(1→4)-D-Glc*p*- (five residues, **LNF**); experimental data: Reference [31]γ-cyclodextrin (eight residues, **GCY**); experimental data: Reference [32]

Estimates of ${}^{13}$C-NMR parameters $T_{1}$, $T_{2}$ and the NOE for selected CH and CH${}_{2}$ probes were calculated by SALEM starting from a local-minimum structure (Figure 2, Figure 3, Figure 4, Figure 5, Figure 6 and Figure 7) and the related Hessian matrix. These were obtained through the Tinker 8 program package [27] using the popular MM3 FF [33,34,35] (see References [36,37] for a review of several FFs for carbohydrates). Minimization was performed through the minimize tool of Tinker with an “RMS gradient per atom criterion” of 0.01 Å. The Hessian matrix was calculated through the Tinker utility testhess. In the hydrodynamic model used to evaluate the friction tensor by SALEM, four quantities were required, namely: the temperature, the local viscosity, the hydrodynamic boundary conditions and the effective radius of the atoms, $R_{\mathrm{eff}}$. Temperature and viscosity were set by the experimental conditions. For sugars in water or polar solvents, stick boundary conditions can be considered appropriate. The only true free parameter was the effective radius.

For each system, the optimal $R_{\mathrm{eff}}$ parameter was determined as the one providing the lowest sum of squared percentage deviations for $T_{1}$, $T_{2}$ and the NOE over the entire ensemble of experimental data available for that system. In Table 1, Table 2, Table 3, Table 4, Table 5 and Table 6, for each set of available experimental data (typeset in boldface in the tables), the optimal theoretical estimates are reported for each system together with the value of the associated optimal $R_{\mathrm{eff}}$ and with the percentage deviations from the experimental $T_{1}$, $T_{2}$, and the NOE (e($T_{1}$), e($T_{2}$), e(NOE), respectively).

The analysis of the various oligosaccharides studied in this work allowed us to attempt a comparative analysis of the model performance and sensitivity to some experimental parameters, with the caveat that a systematic investigation, which goes beyond the preliminary nature of this work, should be performed. The most important factor is temperature. Increasing the temperature implies lowering the viscosity, and thus increasing the principal values of the diffusion tensor. Assuming that in the range of temperatures at which the experiments were carried out, the conformational free energy of the molecules is not changed, increasing the diffusion tensor caused a decrease of the characteristic correlation times, which became closer to the extreme narrowing limit condition. This, in turn, had the effect of increasing $T_{1}$ and $T_{2}$, as well as making them much closer to each other (since the rotational anisotropy is low in small molecules). Reaching the extreme narrowing limit, the spectral densities tended to become proportional to the overall tumbling correlation time, thus losing sensitivity on the conformational dynamics. Therefore, a worse performance of the model was observed and should not be unexpected at lower temperatures/larger viscosities. In the case of **BGL** at 253 K (Table 2), which is the case of a lower temperature and a solvent with higher viscosity, we found an agreement with experimental data within 20% of the relative error, which was the worst scenario in all the test calculations presented here. Simulations around 298 K and in solvents with a similar viscosity showed an agreement within 10% with the experiments.

The relaxation data of the penta-saccharide LNF were calculated on five probes located on the five different sugar rings. At fixed temperature and viscosity conditions, NMR relaxation depends on local geometry (i.e., how the C-H probes are oriented with respect to the diffusion tensor principal axes) and on the conformational free energy. In [38], the authors studied the effect of the shape of the **R2R** Potential Energy Surface (PES) along the Ψ angle on $T_{1}$, $T_{2}$, and the NOE. From MD simulations, it was found that the PES was bistable. The calculation of the NMR relaxation data with the harmonic approximation around the most important minimum led to a 10% error with respect to the calculation done with the bistable potential. The reader is encouraged to inspect Figure 5 of [31]. It reported the 2D PESs along the four (Φ, Ψ) couples of dihedral angles connecting the different sugar units. The PES along the angles connecting rings C and D was bistable, while the other three surfaces could be considered, as first approximation, harmonic. A one-to-one mapping between the probe position and the conformational free energy was not possible. Globally, the approximations done in the SFB model implied that the estimation of a few NMR relaxation data was outside the experimental error (e.g., $T_{1}$ for the A and the E rings), and the overall agreement was 5–10% worse than the agreement found with the diffusive chain stochastic model applied in the past [31].

The sensitivity of the method at different spectrometer frequencies can be also discussed. Even if, again, a trend cannot be strictly highlighted (since there were many factors playing at the same time: geometry, PES, temperature/viscosity, experimental setup), the simulations at lower frequencies tended to be more accurate than those at higher frequencies. This observation can be rationalized, partially, as follows: by increasing the Larmor frequency, NMR relaxation data were more affected by the tails of the spectral densities, which were in turn more sensitive to the fast-relaxing processes, i.e., internal motions described by the simplified harmonic PES in the SFB model.

Overall, the SFB model tended to perform better at higher temperatures, lower viscosities and lower spectrometer frequencies and for molecules with a limited internal flexibility. However, our purpose here was to propose the SFB approach as a general tool, with minimal parametrization, capable of interpreting diverse experimental observations without resorting to ad hoc hypotheses concerning the molecular geometry, internal PES, dissipative properties and so on. The cases presented in this study showed that in the best conditions, relative errors within 5% were found, which usually were compatible with experimental errors. The average percentage deviation over all calculations was 8.5, 5.8 and 7.7 for $T_{1}$, $T_{2}$ and the NOE, respectively. The maximum percentage deviation was 22.4, 13.6 and 18.5, respectively. Such results are significant, if one considers the relative drastic approximations included in the SFB model, confirming—at least for this class of systems—its performative capabilities, which are amenable to be considerably improved by lifting some approximations, like the harmonic nature of the internal PES and the upgrade of the estimates of the friction tensor to include hydrophilicity/hydrophobicity effects. The latter notation was based on the observed estimates of the only free parameter of the model, the effective radius $R_{\mathrm{eff}}$. The optimal $R_{\mathrm{eff}}$ value was found in the range 1.6–2.2 Å for all systems except **LNF**, for which a higher value of 3.2 Å was obtained, which could be due to the molecule being particularly hydrophilic. This, in turn, implies that the molecular dimensions should be increased to take into account a layer of water surrounding the penta-saccharide.

## 4. Discussion

The results reported in the previous section showed that the SFB model reproduced the observed relaxation times and the NOEs of the set of oligosaccharides reported here, with an average accuracy of 5–10%, thus proving the ability of capturing the long-range dynamics of these systems. The study took into account molecules of increasing size, and there was no significant drift in the performance of the model with increasing molecular dimensions. Such an observation is promising, suggesting that the model could profitably be employed for large macromolecules, provided they can be still described as semi-flexible objects, i.e., molecules that mainly fluctuate about a minimum free energy structure (e.g., globular proteins).

The agreement with experimental data was in most cases within 10% relative error. In some cases, higher errors up to 22% were observed. Such discrepancies were expected since free energy profiles along sugars’ Φ, Ψ tetrahedral angles exhibited bistable energy profiles, while only harmonic energy profiles were used in SALEM. In the last case analysed here, **GCY**, as discussed in [32], the rotation of the hydroxymethyl group with respect to the sugar ring described by the torsional angle θ—see Figure 7— featured two energy minima at θ≈−70 and θ≈+50, with the former conformation being the predominant one. The Tinker energy minimization led to a structure with six over eight units in the predominant conformation, and the results shown above were those obtained running SALEM on one of these six probes.

Still, the present general purpose SFB model showed a good performance if compared with straightforward results obtained via molecular dynamics simulations, especially considering that O(μs) trajectories were required in the latter case [28,39]. As mentioned before, the only free parameter here was $R_{eff}$, which was adapted case by case, but as a rule of thumb, a starting value of 2 Å usually provided good agreement with the experiment.

### Perspectives

The main purpose of this paper was to validate, for a class of similar molecular systems for which we have directly controllable published NMR relaxation data, a general stochastic model based on a minimal amount of assumed phenomenological information. In our previous work [18,19], we formulated a systematic approach to describe the dynamics of a non-rigid molecule, based on elaborations from fundamental classical and statistical mechanics, in the form of a family of multidimensional Fokker–Planck operators for the probability density of internal and external degrees of freedom, retaining inertial effects and dissipation. The SFB model is the simplest implementation of this general methodology [19]. This approach seemed particularly relevant for the description of large molecular objects, such as proteins, which represent the main domain of application in the authors’ perspective. The method provides a physically sound framework and is amenable to an efficient treatment at a modest computational price. We are currently working along three possible lines of development: (i) first of all, we are applying the present implementation of the SFB approach for interpreting NMR relaxation data of medium-sized folded proteins in solution, without additional improvement, barring a more accurate evaluation of the internal PES; (ii) at the same time, we are also exploring the effects of including large amplitude motions in the SFB model to account for locally more mobile regions; finally, (iii) we are streamlining SALEM, to make its usage as clear as possible with intuitive interfaces and instructions, to release it as a free tool for the general community. 

## Figures and Tables

**Figure 1 molecules-26-02418-f001:**
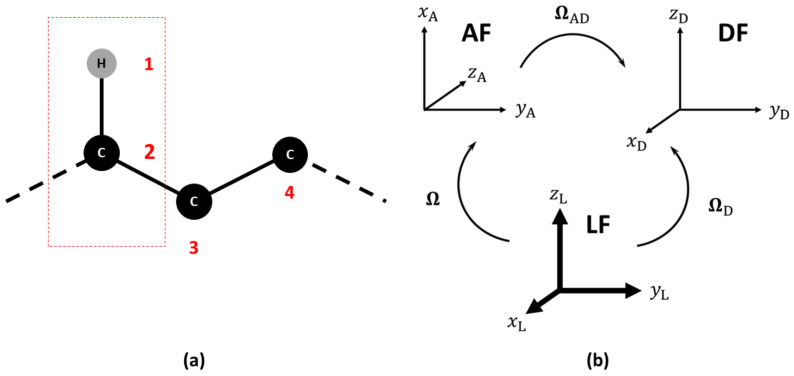
(**a**) Choice of the reference atoms in the calculation of NMR relaxation observables. (**b**) Relevant reference frames: Laboratory (LF), Molecular (AF) and Dipolar Frames (DF). The sets of Euler angles to transform among the frames are also shown. The sets Ω and $\Omega_{D}$ are time dependent, while $\Omega_{\mathrm{AD}}$ is time independent.

**Figure 2 molecules-26-02418-f002:**
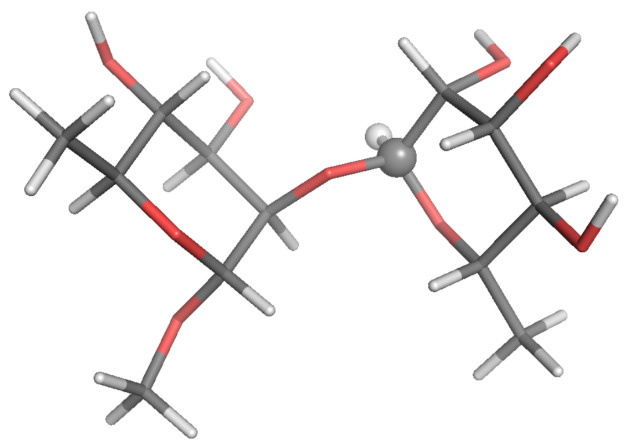
Local-minimum structure of the **R2R** system. Atoms of the active probe are highlighted as spheres.

**Figure 3 molecules-26-02418-f003:**
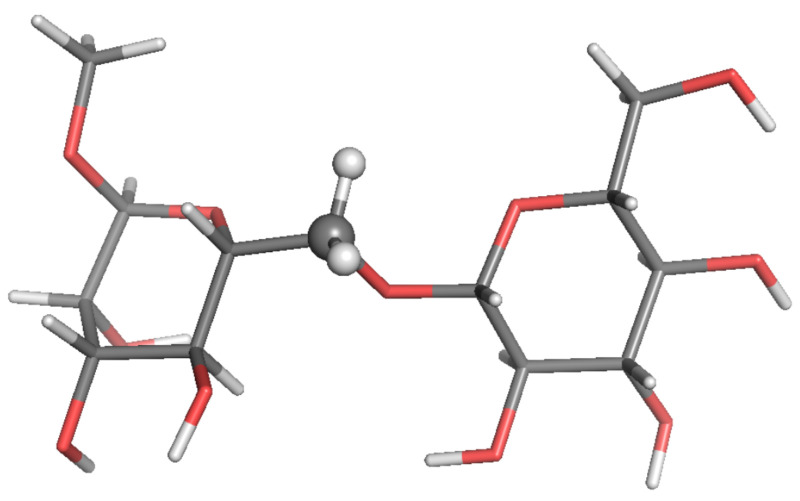
Local-minimum structure of the **BGL** system. Atoms of the active probe are highlighted as spheres.

**Figure 4 molecules-26-02418-f004:**
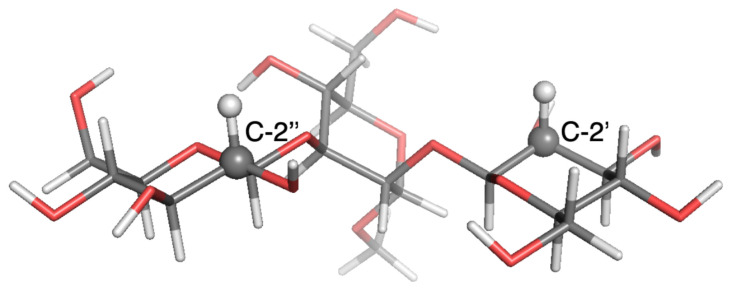
Local-minimum structure of the **GGM** system. Atoms of the active probes are highlighted as spheres.

**Figure 5 molecules-26-02418-f005:**
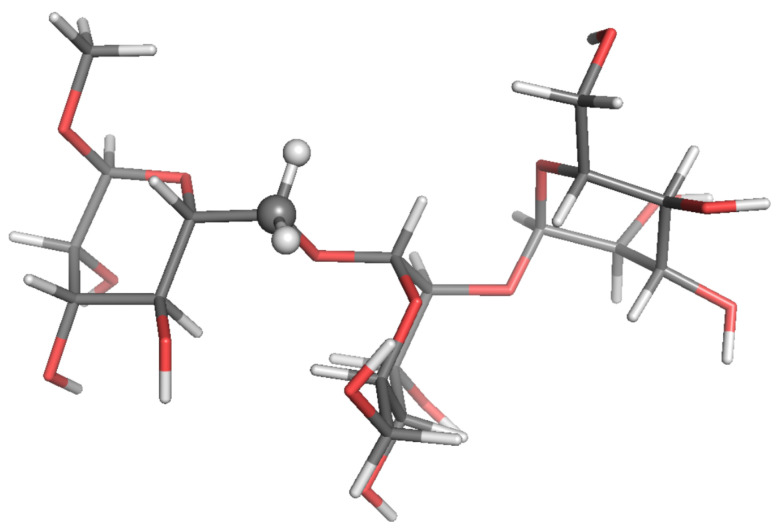
Local-minimum structure of the **TRI** system. Atoms of the active probe are highlighted as spheres.

**Figure 6 molecules-26-02418-f006:**
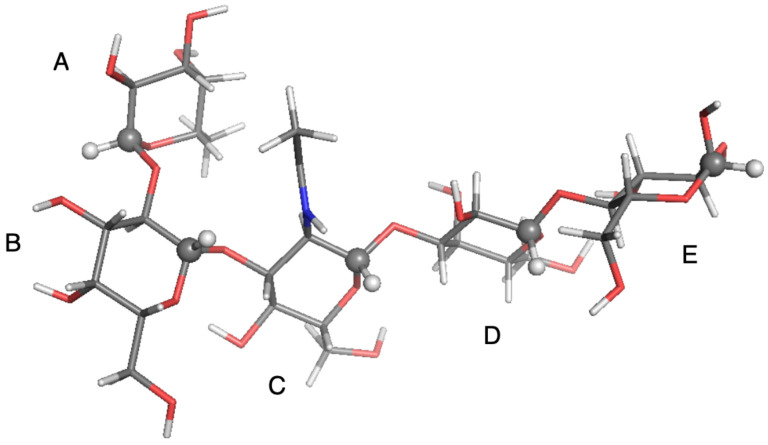
Local-minimum structure of the **LNF** system. Atoms of the active probes are highlighted as spheres.

**Figure 7 molecules-26-02418-f007:**
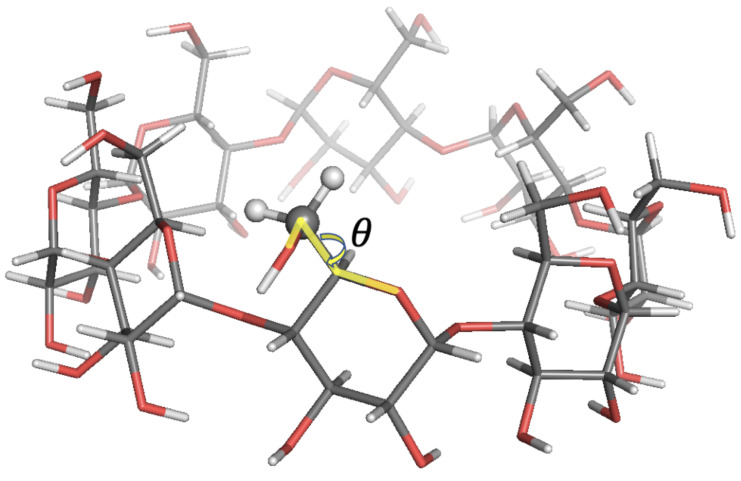
Local-minimum structure of the **GCY** system. Atoms of the active probe are highlighted as spheres. The torsional angle θ for the rotation of the hydroxymethyl group with respect to the sugar ring is also highlighted.

**Table 1 molecules-26-02418-t001:** Experimental and calculated relaxation parameters for system **R2R**. The optimal $R_{\mathrm{eff}}$ and the percentage deviations from the experimental $T_{1}$, $T_{2}$, and the NOE (e ($T_{1}$), e ($T_{2}$), e (NOE), respectively) are also reported.

Probe: ${}^{13}$CH, Solvent: DMSO-*d*${}_{6}$,*T*/K: 298.2, Visc./(Pa s): 2.19 × 10${}^{-3}$								
**Freq./MHz**		$T_{1}$ **/ms**	$T_{2}$ **/ms**	**NOE**	$R_{\mathrm{eff}}$	**e (** $T_{1}$ **)**	**e (** $T_{2}$ **)**	**e (NOE)**
600.1	**exp.**	**440.0**	**402.6**	**2.361**	
	calc.	449.9	420.5	2.308	1.6	2.3	4.4	2.3
700.0	**exp.**	**475.6**	**432.9**	**2.215**	
	calc.	497.5	456.1	2.150	1.6	4.6	5.4	2.9

**Table 2 molecules-26-02418-t002:** Experimental and calculated relaxation parameters for system **BGL**. The optimal $R_{\mathrm{eff}}$ and the percentage deviations from the experimental $T_{1}$, $T_{2}$ and the NOE (e ($T_{1}$), e ($T_{2}$), e (NOE), respectively) are also reported.

Probe: ${}^{13}$CH${}_{2}$, Solvent: DMSO-*d*${}_{6}$/D${}_{2}$O 7:3 Molar Ratio,*T*/K: 253, Visc./(Pa s): 2.82 × 10${}^{-2}$								
**Freq./MHz**		$T_{1}$ **/ms**	$T_{2}$ **/ms**	**NOE**	$R_{\mathrm{eff}}$	**e (** $T_{1}$ **)**	**e (** $T_{2}$ **)**	**e (NOE)**
400	**exp.**	**150**	**32.5**	**1.03**
	calc.	177	31.4	1.22	1.8	17.8	3.2	18.5
600	**exp.**	**284**	**30.1**	**1.08**
	calc.	344	33.1	1.21	1.8	21.1	10.1	12.2
T**/K: 263, Visc./(Pa s): 1.42** × **10**${}^{-2}$								
**Freq./MHz**		$T_{1}$ **/ms**	$T_{2}$ **/ms**	**NOE**	$R_{\mathrm{eff}}$	**e (** $T_{1}$ **)**	**e (** $T_{2}$ **)**	**e (NOE)**
400	**exp.**	**117**	**48.0**	**1.19**
	calc.	121	53.7	1.26	1.8	3.6	11.8	5.9
600	**exp.**	**205**	**55.0**	**1.10**
	calc.	219	61.0	1.23	1.8	6.9	10.9	12.1
T**/K: 293, Visc./(Pa s): 4.30** × **10**${}^{-3}$								
**Freq./MHz**		$T_{1}$ **/ms**	$T_{2}$ **/ms**	**NOE**	$R_{\mathrm{eff}}$	**e (** $T_{1}$ **)**	**e (** $T_{2}$ **)**	**e (NOE)**
400	**exp.**	**127**	**106**	**1.86**
	calc.	124	107	1.78	1.8	2.0	1.0	4.5
600	**exp.**	**166**	**128**	**1.58**
	calc.	170	133	1.51	1.8	2.6	3.8	4.5
900	**exp.**	**219**	**152**	**1.45**
	calc.	249	153	1.33	1.8	13.6	0.6	8.3

**Table 3 molecules-26-02418-t003:** Experimental and calculated relaxation parameters for system **GGM**. The optimal $R_{\mathrm{eff}}$ and the percentage deviations from the experimental $T_{1}$, $T_{2}$ and the NOE (e ($T_{1}$), e ($T_{2}$), e (NOE), respectively) are also reported.

Solvent: D${}_{2}$O,*T*/K: 298.6, Visc./(Pa s): 1.09 × 10${}^{-3}$, Probe:${}^{13}$CH on C-2′								
**Freq./MHz**		$T_{1}$ **/ms**	$T_{2}$ **/ms**	**NOE**	$R_{\mathrm{eff}}$	**e (** $T_{1}$ **)**	**e (** $T_{2}$ **)**	**e (NOE)**
600.13	**exp.**	**456.2**	**416.6**	**2.398**
	calc.	453.2	425.7	2.299	1.8	0.7	2.2	4.1
699.87	**exp.**	**491.1**	**447.4**	**2.267**
	calc.	503.8	463.6	2.148	1.8	2.6	3.6	5.3
**Probe:** ${}^{13}$ **CH on C-2′**								
**Freq./MHz**		$T_{1}$ **/ms**	$T_{2}$ **/ms**	**NOE**	$R_{\mathrm{eff}}$	**e (** $T_{1}$ **)**	**e (** $T_{2}$ **)**	**e (NOE)**
600.13	**exp.**	**491.6**	**450.2**	**2.466**
	calc.	470.0	444.6	2.358	1.8	4.4	1.2	4.4
699.87	**exp.**	**524.5**	**483.9**	**2.346**
	calc.	521.5	483.6	2.203	1.8	0.6	0.1	6.1

**Table 4 molecules-26-02418-t004:** Experimental and calculated relaxation parameters for system **TRI**. The optimal $R_{\mathrm{eff}}$ and the percentage deviations from the experimental $T_{1}$, $T_{2}$ and the NOE (e ($T_{1}$), e ($T_{2}$), e (NOE), respectively) are also reported.

Probe: ${}^{13}$CH${}_{2}$, Solvent: DMSO-*d*${}_{6}$/D${}_{2}$O 7:3 Molar Ratio,*T*/K: 298, Visc./(Pa s): 3.66 × 10${}^{-3}$								
**Freq./MHz**		$T_{1}$ **/ms**	$T_{2}$ **/ms**	**NOE**	$R_{\mathrm{eff}}$	**e (** $T_{1}$ **)**	**e (** $T_{2}$ **)**	**e (NOE)**
500	**exp.**	**144.79**	**111.57**	**1.670**
	calc.	143.60	108.21	1.481	2.2	0.8	3.0	11.3
600	**exp.**	**167.59**	**117.67**	**1.460**
	calc.	169.40	118.20	1.422	2.2	1.1	0.4	2.6
700	**exp.**	**188.36**	**124.55**	**1.320**
	calc.	197.78	127.06	1.385	2.2	5.0	2.0	4.9

**Table 5 molecules-26-02418-t005:** Experimental and calculated relaxation parameters for system **LNF**. The optimal $R_{\mathrm{eff}}$ and the percentage deviations from the experimental $T_{1}$, $T_{2}$ and the NOE (e ($T_{1}$), e ($T_{2}$), e (NOE), respectively) are also reported.

Solvent: DMSO-*d*${}_{6}$/D_2_O 7:3 Molar Ratio,*T*/K: 303, Visc./(Pa s): 1.40 × 10${}^{-3}$, Probe:^13^CH on C-1 (A Residue)								
**Freq./MHz**		$T_{1}$ **/ms**	$T_{2}$ **/ms**	**NOE**	$R_{\mathrm{eff}}$	**e (** $T_{1}$ **)**	**e (** $T_{2}$ **)**	**e (NOE)**
600	**exp.**	**305.0**	**222.0**	**1.460**
	calc.	349.5	213.0	1.489	3.2	14.6	4.1	2.0
700	**exp.**	**354.0**	**244.0**	**1.420**
	calc.	418.6	227.9	1.451	3.2	18.2	6.6	2.2
**Probe:** ${}^{13}$ **CH on C-1 (B residue)**								
**Freq./MHz**		$T_{1}$ **/ms**	$T_{2}$ **/ms**	**NOE**	$R_{\mathrm{eff}}$	**e (** $T_{1}$ **)**	**e (** $T_{2}$ **)**	**e (NOE)**
600	**exp.**	**319.0**	**241.0**	**1.600**
	calc.	336.3	214.4	1.423	3.2	5.4	11.0	11.0
700	**exp.**	**366.0**	**264.0**	**1.530**
	calc.	400.3	229.9	1.386	3.2	9.4	12.9	9.4
**Probe:** ${}^{13}$ **CH on C-1 (C residue)**								
**Freq./MHz**		$T_{1}$ **/ms**	$T_{2}$ **/ms**	**NOE**	$R_{\mathrm{eff}}$	**e (** $T_{1}$ **)**	**e (** $T_{2}$ **)**	**e (NOE)**
600	**exp.**	**318.0**	**225.0**	**1.630**
	calc.	336.6	235.9	1.438	3.2	5.9	4.8	11.8
700	**exp.**	**360.0**	**240.0**	**1.560**
	calc.	395.9	254.1	1.392	3.2	10.0	6.9	10.7
**Probe:** ${}^{13}$ **CH on C-1 (D residue)**								
**Freq./MHz**		$T_{1}$ **/ms**	$T_{2}$ **/ms**	**NOE**	$R_{\mathrm{eff}}$	**e (** $T_{1}$ **)**	**e (** $T_{2}$ **)**	**e (NOE)**
600	**exp.**	**372.0**	**286.0**	**1.740**
	calc.	365.8	275.2	1.525	3.2	1.7	3.8	12.3
700	**exp.**	**404.0**	**302.0**	**1.690**
	calc.	421.2	296.7	1.469	3.2	4.3	1.7	13.1
**Probe:** ${}^{13}$ **CH on C-1 (E residue)**								
**Freq./MHz**		$T_{1}$ **/ms**	$T_{2}$ **/ms**	**NOE**	$R_{\mathrm{eff}}$	**e (** $T_{1}$ **)**	**e (** $T_{2}$ **)**	**e (NOE)**
600	**exp.**	**325.0**	**259.0**	**1.490**
	calc.	390.1	270.1	1.728	3.2	20.0	4.3	15.6
700	**exp.**	**374.0**	**262.0**	**1.500**
	calc.	455.5	289.9	1.677	3.2	21.8	10.6	11.8

**Table 6 molecules-26-02418-t006:** Experimental and calculated relaxation parameters for system **GCY**. The optimal $R_{\mathrm{eff}}$ and the percentage deviations from the experimental $T_{1}$, $T_{2}$ and the NOE (e ($T_{1}$), e ($T_{2}$), e (NOE), respectively) are also reported.

Probe: ${}^{13}$CH${}_{2}$, Solvent: DMSO-*d*${}_{6}$/D${}_{2}$O 7:3 Molar Ratio,*T*/K: 323, Visc./(Pa s): 2.90 × 10${}^{-3}$								
**Freq./MHz**		$T_{1}$ **/ms**	$T_{2}$ **/ms**	**NOE**	$R_{\mathrm{eff}}$	**e (** $T_{1}$ **)**	**e (** $T_{2}$ **)**	**e (NOE)**
400	**exp.**	**123.5**	**75.00**	**1.430**
	calc.	130.7	81.59	1.458	1.8	5.9	8.8	1.9
600	**exp.**	**187.0**	**85.00**	**1.330**
	calc.	213.3	96.58	1.400	1.8	14.1	13.6	5.3
900	**exp.**	**314.5**	**110.9**	**1.250**
	calc.	370.3	109.4	1.384	1.8	17.7	1.3	10.7
T**/K: 343, Visc./(Pa s): 2.30** × **10**${}^{-3}$								
**Freq./MHz**		$T_{1}$ **/ms**	$T_{2}$ **/ms**	**NOE**	$R_{\mathrm{eff}}$	**e (** $T_{1}$ **)**	**e (** $T_{2}$ **)**	**e (NOE)**
400	**exp.**	**134.1**	**107.2**	**1.630**
	calc.	132.5	96.8	1.498	1.8	1.1	9.7	8.1
600	**exp.**	**183.8**	**130.1**	**1.510**
	calc.	203.6	117.3	1.405	1.8	10.8	9.9	6.9
900	**exp.**	**274.0**	**154.8**	**1.330**
	calc.	335.3	137.0	1.366	1.8	22.4	11.5	2.7

## Data Availability

Data sharing not applicable.

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
