# Peer review of "Stochastic Modelling of 13C NMR Spin Relaxation Experiments in Oligosaccharides"

_molecules, 2021, doi:10.3390/molecules26092418_

Round 1

Reviewer 1 Report

A new approach concerning stochastic modelling of 13C NMR spin relaxation experiments in oligosaccharides has been presented. As models, the authors applied a series of six polysaccharides, in which they selected C-H atom pairs. Then, they have calculated the parameters dependent on molecular motion and interaction between atoms, namely, T1 and T2 relaxation time and NOE enhancement. They compared the computed values with available experimental findings, obtaining good agreement between them. Presented investigations are parts of a series of works on a similar subject. I recommend publishing this work as it is, without changes.

Author Response

A new approach concerning stochastic modelling of 13C NMR spin relaxation experiments in oligosaccharides has been presented. As models, the authors applied a series of six polysaccharides, in which they selected C-H atom pairs. Then, they have calculated the parameters dependent on molecular motion and interaction between atoms, namely, T1 and T2 relaxation time and NOE enhancement. They compared the computed values with available experimental findings, obtaining good agreement between them. Presented investigations are parts of a series of works on a similar subject. I recommend publishing this work as it is, without changes.

We thank the reviewer for her/his appreciation of our work.

Reviewer 2 Report

This article described an investigation of the SFB performance to the evaluation of 13C NMR relaxation parameters, T1 and T2, and heteronuclear NOE of several oligosaccharides, which were previously interpreted on the basis of refined ad hoc modelling.  The computational strategy and implementation of the method were discussed, suggesting that the proposed approach can be safely applied to diverse classes of molecular systems.  The present manuscript should be considered for revision, referring to the following comments:

Comments:

  • Regarding the differences between experimental and calculated NMR relaxation parameters, it seems that there are some tendencies: In the BGL system (Fig. 3), differences decreased at the higher temperature (293 K). In the GGM and TRI systems, NMR relaxation parameters were evaluated at 298.6 and 298 K, respectively, indicating that the relatively small differences were observed.  How is the effect of temperature?

  • In the LNF system (Fig. 6), dependence of the residue positions should be discussed. Did the residue positions affect the calculated results?

  • In the GCY system (Fig. 7), the parameters using three NMR frequencies, such as 400, 600 and 900, were evaluated. How was the frequency dependence?  The effects of viscosity of solvents should also be described. 

  • In the present paper, oligosaccharides were selected as the model compounds. Have the authors evaluated the NMR parameters for peptides or proteins?  Because many relaxation studies were carried out in peptides and proteins, this reviewer is interested in the achievement toward them.

Author Response

This article described an investigation of the SFB performance to the evaluation of 13C NMR relaxation parameters, T1 and T2, and heteronuclear NOE of several oligosaccharides, which were previously interpreted on the basis of refined ad hoc modelling.  The computational strategy and implementation of the method were discussed, suggesting that the proposed approach can be safely applied to diverse classes of molecular systems.  The present manuscript should be considered for revision, referring to the following comments:

Comments:

  • Regarding the differences between experimental and calculated NMR relaxation parameters, it seems that there are some tendencies: In the BGL system (Fig. 3), differences decreased at the higher temperature (293 K). In the GGM and TRI systems, NMR relaxation parameters were evaluated at 298.6 and 298 K, respectively, indicating that the relatively small differences were observed.  How is the effect of temperature?
  • In the LNF system (Fig. 6), dependence of the residue positions should be discussed. Did the residue positions affect the calculated results?
  • In the GCY system (Fig. 7), the parameters using three NMR frequencies, such as 400, 600 and 900, were evaluated. How was the frequency dependence?  The effects of viscosity of solvents should also be described. 

We thank the reviewer for her/his comments. We have extended the Results section of the paper with a short discussion on the accuracy of the SFB model with respect to experimental conditions (temperature, viscosity, spectrometer frequency). As the reviewer noticed, there are some weak trends that emerge by looking at the calculations. Finding and discussing trends is difficult since the set of oligosaccharides used for the calculations was not intended for the purpose of systematically analysing how the model performs changing each single parameter while keeping the others fixed. However, following the Reviewer’s suggestions, we were able to provide a tentative rationalization on the accuracy of the SFB approach with respect to the above-mentioned parameters.

In the present paper, oligosaccharides were selected as the model compounds. Have the authors evaluated the NMR parameters for peptides or proteins?  Because many relaxation studies were carried out in peptides and proteins, this reviewer is interested in the achievement toward them.

We hope to present soon a systematic exploration addressing the treatment of proteins. We would like to stress that our paper does not intend to propose a systematic analysis of trends of accuracy and performances of the model, but rather to show preliminarily its capability to interpret significant spectroscopic signatures of a class of similar molecules, for which we have published data. As we now write in the Discussion, we are working of an extensive analysis of the model, from three different points of view: 1) investigating its application to larger molecular systems (proteins and  polysaccharides) , 2) describing in details the applicative code SALEM, which is going to be released as an open source tool and 3) upgrading some approximations, namely the inclusion of hydrophilicity/hydrophobicity effects, better description of internal PES and introduction of activated conformational processes.

Reviewer 3 Report

This manuscript describes calculations of a couple of common NMR parameters (T1, T2, NOE), using a methodology that can be found in the references (18 and 19), for some concrete chemicals, and compare them with numbers, that are found in the literature (references 27-31). The agreement is within about 22%. There is only one adjustable parameter: Reff (the effective radius of the atoms).

The calculations rely on the integrated software package SALEM, that is not yet available to the scientific community, as the authors acknowledged (lines 248-249). However that restricts the usefulness of this paper as the reviewer and reader would find it difficult to reproduce the results of this paper, as well as to apply the method to their own chemicals of interest. Thus, this paper is more of a exploratory contribution than a real useful guide.

Still, with the preview of the SALEM program in lines 246-261, the approach of this manuscript appears promising. The impact of this manuscript would have been much greater, if the SALEM program had already been released or described in more detail. Presumably, the authors want to describe the SALEM program in a future manuscript. However, I would advise the authors to include a more detailed description of the SALEM program, or at least some more details of the calculations, in this manuscript. The question that needs to be answered is: How can someone do the calculations without having access to the SALEM program?

Some specific issues:
Line 7: Put "SFB" in parenthesis.
Line 17: Better substitute: "The macroscopic physical observables" => "Some of the macroscopic physical observables" as they may be other physical observables that are not mentioned.
Line 100: "choise" => "choice"
Line 224: "reasonable accuracy": Please quantify the accuracy for all data (difference between exp. and calc.) and update the tables.
Tables associated with Figures 3-7): Please label the appropriate row as "exp." or "calc.".

Author Response

This manuscript describes calculations of a couple of common NMR parameters (T1, T2, NOE), using a methodology that can be found in the references (18 and 19), for some concrete chemicals, and compare them with numbers, that are found in the literature (references 27-31). The agreement is within about 22%. There is only one adjustable parameter: Reff (the effective radius of the atoms).

The calculations rely on the integrated software package SALEM, that is not yet available to the scientific community, as the authors acknowledged (lines 248-249). However, that restricts the usefulness of this paper as the reviewer and reader would find it difficult to reproduce the results of this paper, as well as to apply the method to their own chemicals of interest. Thus, this paper is more of an exploratory contribution than a real useful guide.

Still, with the preview of the SALEM program in lines 246-261, the approach of this manuscript appears promising. The impact of this manuscript would have been much greater if the SALEM program had already been released or described in more detail. Presumably, the authors want to describe the SALEM program in a future manuscript. However, I would advise the authors to include a more detailed description of the SALEM program, or at least some more details of the calculations, in this manuscript. The question that needs to be answered is: How can someone do the calculations without having access to the SALEM program?

We recognize the merit of the Reviewer’s comments. Indeed, we are aware that assuring the availability of SALEM to the general audience would strengthen the impact of the paper. However, our limited purpose in this paper is only to show the main capabilities of this approach, and we are working to make public, under an open-source license, our code, with suitable instructions and an efficient GUI for an easy use. We have partially rearranged our text at the end of the “Parameterization and implementation” subsection to better clarify this point.  Indeed, as we now write in the Discussion, we are working of an extensive analysis and presentation of the model, from three different points of view: 1) investigating its application to larger molecular systems (proteins and polysaccharides), 2) describing in detail the applicative code SALEM and 3) upgrading some approximations, namely the inclusion of hydrophilicity/hydrophobicity effects, better description of internal PES and introduction of activated conformational processes.

Some specific issues:

  • Line 7: Put "SFB" in parenthesis.
  • Line 17: Better substitute: "The macroscopic physical observables" => "Some of the macroscopic physical observables" as they may be other physical observables that are not mentioned.
    Line 100: "choise" => "choice"
  • Line 224: "reasonable accuracy": Please quantify the accuracy for all data (difference between exp. and calc.) and update the tables.
  • Tables associated with Figures 3-7): Please label the appropriate row as "exp." or "calc.".

All the reviewer’s suggestions have been implemented. Additionally, a general revision of the text has been performed to eliminate residual typos and misprints. In the main text we quantify specifically the percentage average and maximum error. The tables have rows labelled according to being referred to experimental or calculated values, with the errors (differences between experimental and calculated values) in adjacent columns.

Round 2

Reviewer 3 Report

The comments of this reviewer have been addressed by the authors. The readers and this reviewer is looking forward to the promised extensive analysis and presentation of the SALEM model in the near future.